# Functional Mechanisms and the Application of Developmental Regulators for Improving Genetic Transformation in Plants

**DOI:** 10.3390/plants13202841

**Published:** 2024-10-10

**Authors:** Yilin Jiang, Siyuan Liu, Xueli An

**Affiliations:** 1Research Institute of Biology and Agriculture, Zhongzhi International Institute of Agricultural Biosciences, School of Chemistry and Biological Engineering, University of Science and Technology Beijing, Beijing 100083, China; b20190393@xs.ustb.edu.cn (Y.J.); m202321057@xs.ustb.edu.cn (S.L.); 2Beijing Engineering Laboratory of Main Crop Bio-Tech Breeding, Beijing International Science and Technology Cooperation Base of Bio-Tech Breeding, Beijing Solidwill Sci-Tech Co., Ltd., Beijing 100192, China

**Keywords:** developmental regulators, regulatory mechanisms, plant regeneration, genetic transformation efficiency, crop improvement

## Abstract

Enhancing the genetic transformation efficiency of major crops remains a significant challenge, primarily due to their suboptimal regeneration efficiency. Developmental regulators, known as key regulatory genes, involved in plant meristem and somatic embryo formation, play a crucial role in improving plant meristem induction and regeneration. This review provides a detailed summary of the molecular mechanisms and regulatory networks of many developmental regulators, in the context of enhancing the genetic transformation efficiency in major crops. We also propose strategies for exploring and utilizing additional developmental regulators. Further investigation into the mechanisms of these regulators will deepen our understanding of the regenerative capacity and genetic transformation processes of plants, offering valuable support for future crop improvement efforts. The discovery of novel developmental regulators is expected to further advance crop transformation and the effective manipulation of various developmental regulators could provide a promising approach in order to enhance genetic transformation efficiency.

## 1. Introduction

With the growing global population and the increasing pressure from climate change on agricultural production, traditional plant breeding methods are becoming inadequate to meet the demands of food security and sustainable development [1,2]. In these circumstances, plant genetic transformation technology has become an indispensable tool in modern agriculture, offering a wide range of potential applications. Efficient genetic transformation techniques are fundamental to gene-editing technologies (such as CRISPR/Cas9), which enable precise genome modifications, and to biotechnological breeding methods (such as transgenic breeding) [3,4], thereby significantly accelerating the plant breeding process. Moreover, gene function analysis, including both forward and reverse genetics, also relies on genetic transformation techniques to clarify the roles of specific genes in plant growth and development [5,6,7,8,9,10]. Stable and efficient transformation systems are crucial for both gene knockout and overexpression studies [11,12].

Beyond accelerating plant breeding and elucidating gene function, genetic transformation technology also plays a critical role in ensuring the safety and regulatory compliance of gene-edited and transgenic crops [3,13]. For instance, precise transformation techniques are essential to minimize off-target effects and unintended genetic modifications [14], thereby enhancing the safety and acceptance of genetically modified organisms (GMOs) and gene-edited plants [3]. Genetic transformation techniques further support the development of biosafety management strategies, such as better control of gene flow, the monitoring of environmental impacts, and the assessment of potential risks to human health. Additionally, transformation methods are indispensable for optimizing trait stacking, which involves the simultaneous introduction of multiple desirable genes into plants to enhance complex traits, such as drought tolerance, pest resistance, and nutritional quality [3]. This multifaceted application underscores the importance of genetic transformation technology not only in crop improvement, but also in addressing the challenges of sustainable agriculture in a changing global environment [15].

However, current plant genetic transformation technologies face significant challenges, such as strong genotype dependency, low regeneration efficiency, and the complexity of the transformation process, particularly in crops that are difficult to transform. To address these challenges, researchers have successfully developed various developmental regulators (DRs) to promote plant regeneration and enhance transformation efficiency. These DRs include SOMATIC EMBRYOGENESIS RECEPTOR-LIKE KINASE (SERK) [16,17], LEAFY COTYLEDON (LEC) [18,19,20], WOUND-INDUCED DEDIFFERENTIATION 1 (WIND1) [20,21,22], BABY BOOM (BBM) [23,24,25,26], WUSCHEL (WUS) [27,28,29,30,31,32], the growth-regulating factor–growth regulator interacting factor (GRF–GIF) complex [33,34,35,36], the WUSCHEL-RELATED HOMEOBOX (WOX) [37,38,39,40,41,42], rZmGOLDEN2 [43], DNA binding with One Finger (DOF) [44], and REGENERATION FACTOR 1 (REF1) [45]. By modulating the expression of these DRs, significant improvements in the formation efficiency of meristems and somatic embryos have been achieved, thereby increasing the success rate of genetic transformation. However, the molecular mechanisms according to which these DRs function across different species and genotypes are not yet fully understood, which limits their broader application in agricultural production.

In this review, we focus on summarizing and analyzing the latest progress in the application of DRs in terms of plant genetic transformation, elucidating their roles in enhancing genetic transformation efficiency and exploring their potential application values across different plants. Additionally, we discuss strategies to optimize the combination and application of these DRs to overcome existing bottlenecks in genetic transformation technologies and provide new directions for future research.

## 2. Developmental Regulators for Plant Genetic Transformation: Mechanisms, Advances, and Applications

### 2.1. WUS and BBM Promote Genetic Transformation in Plants

*WUSCHEL* (*WUS*) and *BABY BOOM* (*BBM*) are key regulatory factors of cellular totipotency and they can promote cell dedifferentiation and somatic embryogenesis, which have been corroborated across a variety of plant species, including *Arabidopsis* (*AtWUS* and *AtBBM*) [29,30,46], maize (*Zea mays* L.) (*ZmWUS2* and *ZmBBM*) [31], and rice (*Oryza sativa*) (*OsBBM*) [31]. *OsWUS* is essential for rice tiller development, as it regulates apical dominance and hormonal responses that influence tiller bud outgrowth [47]. *WUS* promotes the integration of exogenous DNA into receptor genomes by maintaining plant receptor cells in an undifferentiated state [48], while *BBM* triggers somatic embryogenesis and facilitates cell proliferation and division [49]. The expression of both genes can significantly enhance the genetic transformation efficiency of difficult-to-transform plant varieties and promote the formation of callus tissue and the regeneration of transformed plants [31].

The synthesis and signaling of plant hormones, particularly auxins and cytokinins, are crucial for maintaining cell fate and organ formation [50]. The functional mechanisms of *WUS* and *BBM* are intricately linked to these hormone signaling networks. Both WUS and BBM transcriptionally regulate *LEAFY COTYLEDON1* (*LEC1*), *LEC2*, and *AGAMOUS-LIKE15* (*AGL15*) to enhance embryonic competence during somatic embryogenesis [49] (Figure 1). *LEC* further modulates downstream hormone signaling pathways. *WUS* and *Arabidopsis Response Regulators* (*ARRs*) positively and negatively regulate cytokinin signaling, respectively [51,52] (Figure 1). The synthesis of auxins, regulated by *Tryptophan Aminotransferase of Arabidopsis 1* (*AtTAA1*), *YUCCA3* (*AtYUC3*), and *AtYUC8*, along with auxin polar transport mediated by *pin-formed 1* (*NtPIN1*) and *NtPIN2*, and signal transduction mainly by *indole-3-acetic acid* (*AtIAA*) and *Auxin Response Factor* (*AtARF*) family members, collectively influence the differentiation process of transformed cells [49,53] (Figure 1). Concurrently, the gibberellin metabolic pathway, regulated by *GA20-oxidase* genes (*AtGA20ox6* and *AtGA3ox2*), and cell proliferation-related factors, such as the *Actin depolymerizing factor* (*AtADF9*) [54,55], play key roles in enhancing genetic transformation efficiency. LEC1/2 (e.g., AtLEC1/2) and MADS-box transcription factor AGAMOUS-LIKE 15 (e.g., AtAGL15) participate in regulating embryonic development and cell proliferation [49], thus constructing a complex regulatory network. The integrated action of *WUS* and *BBM* promotes cell proliferation and shoot formation more efficiently than either gene alone and increases the likelihood of successful transformation events, which can result in the growth of viable transgenic plants. Their synergistic effects greatly increase the genetic transformation efficiency of various plant species, such as maize, rice, and *Arabidopsis,* as well as sorghum (*Sorghum bicolor*), coffee plants, and hemp (*Cannabis sativa*) (Figure 1).

The overexpression of *WUS* and *BBM* has been identified as significantly improving the transformation efficiency of various plants. Lowe et al. were the first to overexpress *ZmBBM* and *ZmWUS2* in immature maize embryos, and the transformation efficiencies of several previously deemed difficult-to-transform maize inbred lines were significantly improved (Table 1) [31]. Additionally, they successfully transformed mature seed embryos and leaf segments from different inbred line seedlings with overexpressed *ZmBBM* and *ZmWUS2* and then obtained normal T_0_ plants. This method significantly enhances the genetic transformation efficiency and reduces the genotype dependency in maize [31].

However, the ectopic expression of both *ZmBBM* and *ZmWUS2* adversely affected the growth of regenerated plants [31]. To mitigate these negative impacts, especially on the growth and fertility of transgenic maize plants, Lowe et al. employed the *ZmPLTP* (phospholipid transfer protein gene) promoter and the auxin-inducible *ZmAxig1* promoter to drive *ZmBBM* and *ZmWUS2*, respectively (Table 1) [56]. This strategy enhances the tissue- and temporal-specificity of the expression of both genes and allows maize to be efficiently and rapidly genetically transformed, overcoming specific genotype dependency and bypassing the callus culture stage. Notably, *ZmPLTP* and *Axig1* promoters are active only when using immature embryos (IEs) as explants. This approach rapidly produces a large number of somatic embryos (SEs) that are directly transformed and grown into plants without the callus stage, and the resulting transgenic plants display normal growth and reproductive capabilities [56].

The prolonged expression of certain DRs leads to abnormal seedlings, which thus need to be removed after the transformation process. Wang et al. explored a new strategy by utilizing the inducible Cyclization Recombination Enzyme (CRE) to remove DRs [57]. During this strategy, the promoters of genes encoding inducible heat-shock proteins, such as Hsp17.7 and Hsp26, were used to drive *CRE* expression during the early embryo development stages to remove unnecessary transgenic elements in regeneration plants. This method, combined with improved tissue culture conditions, can achieve higher T_0_ transformation rates and reduce the occurrence of non-transgenic transformants, thereby improving the transformation efficiency of specific maize inbred lines (Table 1). Additionally, it ensures that the transformants do not contain DRs and/or selection markers, thus providing a more efficient tool for precise genetic improvement.

In plant genetic engineering, the ternary vector system, by combining disarmed Ti plasmids, T-DNA binary vectors, and additional helper plasmids [58], enhances the virulence functions, thereby improving the efficiency of the T-DNA transfer into the plant genome. The cooperative action initiated by additional virulence genes in the system further boosts the transformation efficiency [58]. The ternary vector system is designed for broad applicability across different *Agrobacterium* strains, while the transformation efficiency varies when using different strains. Thus, optimization is required for specific plant species and experimental conditions. The ternary vector system, combining CRE-assisted transgene elimination, is commonly applied to further enhance maize transformation efficiency. Specifically, a ternary vector system was designed to integrate the CRISPR/Cas9 genome-editing module with a CRE excision system, driven by a drought-inducible promoter (*Rab17*) [59]. The novel ternary vector system primarily consists of a new dual-component *pGreen*-like vector, *pGreen3*, along with a virulence auxiliary vector, based on *pVS1* and a *pGreen3* replication auxiliary vector. In conjunction with the DR module, this ternary vector system significantly enhances the *Agrobacterium*-mediated transformation efficiency in maize (Table 1) [59].

**Table 1 plants-13-02841-t001:** Developmental regulators improving genetic transformation in plants.

Developmental Regulators	Encoding Proteins	Genes	Species	Explants	Transformation Effects	Refs.
*BABY BOOM (BBM);* *WUSCHEL (WUS)*	BBM, transcription factors of the AP2/ERF family; WUS, a homeodomain transcription factor	*ZmBBM; ZmWUS2*	*Zea mays*	Immature embryo; mature seed embryo sections; seedling-derived leaf segments	Significantly increases transformation efficiency and reduces genotype dependence for inbred PHH5G, PHP38, PHN46, PH581, and PH0AZ	[31]
				Immature embryo	Healthy and fertile plants are regenerated	[56]
					Transgenic plants are free of morphogenic and marker genes	[57]
					Transformation frequencies increase by 20 times in recalcitrant maize inbred line ND88	[59]
				Seedling-derived leaf tissue	Substantially improves leaf transformation for inbred PH1V69 and B104	[32]
			Sorghum	Seedling-derived leaf tissue	The average frequency of T_0_ sorghum (Tx430 variety) plants in the experimental group ranges from 35% to 37%	[32]
			*Eragrostis tef*	Seedling-derived leaf tissue	T_0_ seedlings with healthy shoots and roots were regenerated	[32]
			*Panicum virgatum*			
			*Cenchrus americanus*			
			*Setaria italica*			
			*Secale cereale*			
			*Hordeum vulgare*			
		*ZmWUS2*	*Zea mays*	Immature embryo	Removal of the morphogenic genes and high transformation frequencies for inbred PHW52, PH2KD1, PH28SV, PH4BAH, PH2Y8G, and PH4B9Z	[60]
		*OsBBM; ZmWUS2*	*Oryza sativa*	Calli	Transformation frequencies increase by 13 times for indica variety IRV95	[31]
		*ZmBBM; ZmWUS2*			Transformation frequencies increase by eight times for indica variety IRV95	
		*AtWUS*	*Medicago Sativa*	Leaflets and hairy root segments	The number of embryos produced per *WUS* callus was 76% and 2.3-fold higher than that of the control	[61]
		*HvBBM;* *HvWUS*	*Hordeum vulgare*	Immature embryo	The transformation frequency significantly exceeded the empty vector control by 17.48%	[62]
		*BnBBM*	*Capsicum annuum*	Cotyledon segment	Compared to the control group with no transgenic plantlets, the transformation efficiencies for the experimental groups were 0.6% and 1.1%, respectively	[26]
		*TcBBM*	*Theobroma cacao*	Cotyledon	Somatic embryo production per explant was 29% higher compared to the control tissue	[63]
		*AtWUS*	*Gossypium hirsutum*	Hypocotyl segment	The differentiation rate of 35S/*WUS* transgenics was 47.14% higher than that of the control group	[30]
		*AtWUS*	*Coffea canephora*	Leaf segment	Expression of *WUS* in coffee plants induces callus formation and boosts somatic embryo production by 400%	[64]
		*MdBBM1*	*Malus domestica*	Leaf segment	The transformation efficiencies were 250% to 327% higher compared to the control group	[65]
		*CsWUS4*	*Cannabis Sativa*	Hypocotyls from immature grain	The shoot regeneration frequency increased by 170%	[66]
*WUSCHEL-RELATED HOMEOBOX* (*WOX*)	WUSCHEL-RELATED HOMEOBOX (WOX) family protein	*TaWOX5*	*Triticum aestivum*	Immature embryo	Much higher transformation frequencies (up to 1.9 times) in 29 wheat (*Triticum aestivum*) varieties	[42]
			*Zea mays*		Transformation frequencies of 19–38% in B73 and A188	
		*ZmWox2a*	*Zea mays*		Production of phenotypically normal and fertile regenerants for inbred lines B73 and LH244	[67]
*Growth-regulating factor–growth regulator interacting factor* (*GRF*–*GIF*)	GRF protein with conserved QLQ and WRC domains and GRF-interacting factor	*TaGRF4;* *TaGIF1*	*Triticum aestivum*	Immature embryo	A maximum enhancement of up to 7.8-fold in regeneration efficiency is observed among cultivars, including Kronos, Desert King, Cadenza, Cadenza, and breeding line UC3184	[34]
			*Oryza sativa*	Mature embryos	The regeneration efficiency increased 2.1 times in the variety Kitaake	
		*CsGRF4;* *CsGIF1*	*Citrus*	Epicotyl	The regeneration frequency increased by 370%	
		*ZmGRF5-LIKE1;* *ZmGRF5-LIKE2*	*Zea mays*	Immature embryo	Transformation frequencies increase by three times for inbred line A188	[35]
		*AtGRF5;* *GmGRF5-LIKE*	*Glycine max* L.	Seedling	More shoot development per explant in cultivars Jake and CD215	
		*AtGRF5*	*Beta vulgaris ssp. vulgaris*	Leaf from micropropagated shoots	A significant 6-fold increase in the transformation efficiency	
		*HaGRF5-LIKE*	*Helianthus annuus*	Cotyledon	Promotes transgenic shoot production	
		*ClGRF4;* *ClGIF1*	*Citrullus lanatus*	Cotyledon	The regeneration efficiency increased by 19.64% to 67.27%	[36]
*DOF*	DNA binding with One Finger	*TaDOF5.6* *TaDOF3.4*	*Triticum*	Immature embryo	A maximum enhancement of up to 2.12 times in regeneration efficiency is observed in Fielder, JM22, and Kenong 199	[44]
*GOLDEN2*	Members of the GARP (Golden2, ARR-B, and Psr1) superfamily of transcription factors	*rZmGOLDEN2*	*Zea mays*	Immature embryo	Transformation frequency improvements of 1.75 times for inbred line B104	[43]
			*Oryza sativa*	Calli	Transformation frequency improvements of 1.31–3.44 times for HuaZhan, HuaGuang, DongJing, and ZhongJia 8	
REF1 (REGENERATION FACTOR 1)	The precursor of SlPep, the sole tomato ortholog of the plant elicitor peptide (Pep) family of immunomodulatory peptides	*SISPR9*	Tomato (*Solanum lycopersicum*)	The stem base of 18-day-old tomato seedlings	Increased the transformation efficiency of the wild tomatoes *S. peruvianum* and *S. habrochaites* by 6 to 12 times	[45]
		*GmREF1*	*Glycine max* L.	Cotyledonary nodes	Increased the transformation efficiency of Dongnong-50 by 2–5 times	
		*TaREF1*	*Triticum*	Immature embryo	Increased the transformation efficiency of JM22 by four times	
		*ZmREF1*	*Zea mays*	Immature embryo	Increased the transformation efficiency of B104 by four times	
*LAX1* *(LAX PANICLE 1)*	Non-canonical basic helix–loop–helix transcription factor	*TaLAX1*	*Triticum*	Immature embryo	Significantly increased the transformation efficiency of the wheat varieties Fielder and Chinese Spring	[68]

In addition to their application in major crops, *WUS* and *BBM* have also been utilized in horticultural and medicinal plants (Figure 1; Table 1). For instance, expressing *AtWUS* in coffee plants (*Coffea canephora*) can induce the formation of callus tissue and increase the amount of somatic embryos by 400% [64]. The ectopic expression of the *MdBBM1* gene in apple plants (*Malus domestica Borkh*) leads to enhanced mRNA levels of cell division activators and reduces those of inhibitors associated with hormonal (auxin, cytokinin, and gibberellin) signaling pathways, which together significantly improve the genetic transformation efficiency in apple plants [65]. Additionally, a recent study has shown that overexpressing DR genes, such as *CsGRF3*–*CsGIF1* and *CsWUS4*, in the hypocotyls of immature grains of hemp (*Cannabis sativa*) can significantly enhance shoot regeneration efficiency [66]. In summary, the functional mechanisms of *WUS* and *BBM* and their application in promoting genetic transformation efficiency in multiple plant species, especially in horticultural and medicinal plants, along with a deep understanding of their interaction with plant hormone regulatory networks, propel the development of advanced genetic transformation strategies required for plant genetic engineering.

### 2.2. Overexpression of SERK Enhances Somatic Embryogenesis

SOMATIC EMBRYOGENESIS RECEPTOR-LIKE KINASE (SERK) belongs to the family of Leucine-Rich Repeat Receptor-Like Kinases (LRR-RLKs), characterized by their ability to recognize diverse extracellular ligands and transduce signals into the cell to trigger specific cellular responses [16,69]. In coffee plants, studies on the overexpression of *CcSERK1* have elucidated its mechanism in regulating somatic embryogenesis by promoting auxin biosynthesis through activating key genes, such as *CcAGL15*, *CcWUS*, *EMBRYOMAKER* (*CcEMK*), *CcBBM CcYUC1/4*, and *CcTAA1*, and by promoting auxin transportation and signaling pathways through activating *PIN1* and *PIN4*, which collectively lead to a significant increase in the number of somatic embryos [70] (Figure 2). Furthermore, *SERKs* are also considered as markers for identifying cells with embryogenic potential [71]. For instance, in rice, the suppression of *OsSERK1* expression significantly reduces the bud regeneration rate, whereas its overexpression leads to the opposite result, highlighting its central role in somatic embryogenesis [71] (Figure 2).

### 2.3. WIND Regulates Cell Dedifferentiation in Wound Response

*WOUND-INDUCED DEDIFFERENTIATION 1* (*WIND1*) belongs to the AP2/ERF transcription factor family, formerly known as *RAP2.4* [21,72,73]. WIND1 can directly activate the expression of the *ENHANCER OF SHOOT REGENERATION 1* (*ESR1*), encoding an AP2/ERF transcription factor in *Arabidopsis*, thereby promoting callus formation and bud regeneration [74] (Figure 2). Moreover, WIND1 also regulates cytokinin signaling, through activating *ARR1* and *ARR12,* to promote callus formation at wound sites [75]. The ectopic expression of *WIND1* not only induces callus formation and adventitious bud regeneration in *Arabidopsis,* but also induces callus formation in rapeseed (*Brassica napus*), tomato, and tobacco (*Nicotiana tabacum*), even on hormone-free media [22] (Figure 2). A recent study has demonstrated that the overexpression of ZmWIND1 can significantly increase the callus induction rates of maize inbred lines Xiang 249 and Zheng 58 to 60.22% and 47.85%, respectively. Additionally, the transformation efficiency was elevated to 37.5% in Xiang 249 and 16.56% in Zheng 58 [15]. Notably, the dual induction of *WIND1* and *LEC2* effectively promotes the formation of embryogenic calluses in rapeseed, while the induction of *WIND1* or *LEC2* alone is unable to successfully form a callus [76], offering a new strategy to enhance plant genetic transformation efficiency by combining *WIND1* with other DRs.

**Figure 2 plants-13-02841-f002:**
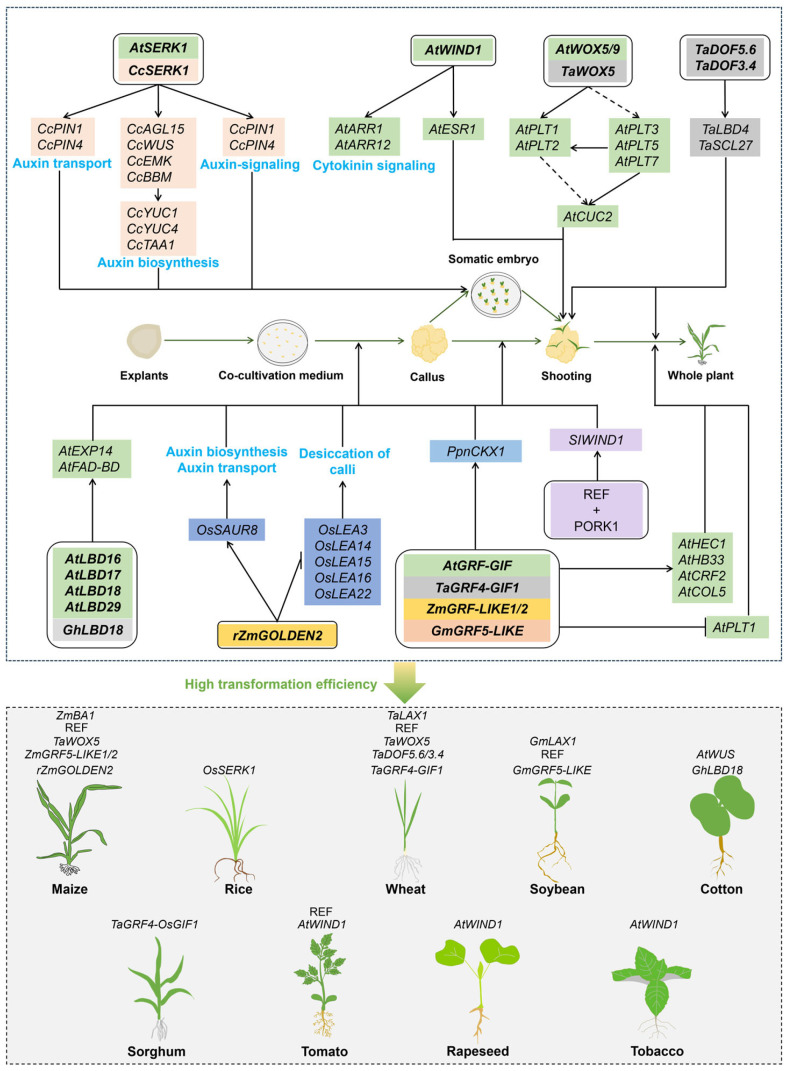
Schematic representation of *SERK*, *WIND1*, *WOX5*, *DOF*, *LBD*, *rZmGOLDEN2*, and *GRF*–*GIF* and REF peptides and their roles in enhancing plant genetic transformation. Genes are color-coded by species: green for *Arabidopsis*, yellow for maize, blue for rice, dark gray for cotton, light gray for wheat, light orange for soybean (*Glycine max*), orange for *Citrus*, light purple for tomato, and light blue for poplar (*Populus*). The light blue font and boxes indicate hormone-related signaling pathways and associated biological processes, as well as their related genes. The genes shown above each plant species represent those employed to improve the genetic transformation efficiency in that particular species. The co-cultivation medium is primarily used for the *Agrobacterium* infection process and is not involved in the callus induction in explants. The processes illustrated reflect maize transformation, as an example.

### 2.4. WOX Involved in Promoting Somatic Embryogenesis

WOX5 is a primary regulatory factor involved in regulating the meristematic activity at root tips and stimulates cell differentiation by recruiting PLETHORA (PLT) factors [77]. Studies in *Arabidopsis* have shown that *PLT3*, *PLT5*, and *PLT7* genes can coordinate to regulate de novo shoot regeneration through activating *PLT1* and *PLT2* genes to promote pluripotency establishment and through regulating *CUP-SHAPED COTYLEDON2* (*AtCUC2*) to complete bud formation [78] (Figure 2). Additionally, studies have shown that *PLT5* can promote in planta transformation in snapdragon (*Antirrhinum majus*) and tomato (*Solanum lycopersicum*), while significantly enhancing shoot regeneration and transformation efficiency in Chinese cabbage (*Brassica rapa*). PLT5 also facilitates the formation of transgenic calli and somatic embryos in sweet pepper (*Capsicum annuum*), through in vitro tissue culture [79].

In addition, *WOX5* enhances callus tissue cell regenerative capacity through its transcriptional activator WRKY DNA-BINDING PROTEIN 23 (WRKY23) and repressor basic helix–loop–helix 041 (bHLH041) in *Arabidopsis* [80], demonstrating its significance in plant regeneration. In tobacco, combinations of *WOX* family members, *WOX2*, *WOX8*, and *WOX9,* promote plant regeneration [81]. Notably, in wheat, the overexpression of *TaWOX5* significantly increases the genetic transformation efficiency [42] (Figure 2, Table 1), thus overcoming the challenge of wheat genotype-dependent transformation. This method is applicable to many commercial and hard-to-transform wheat varieties, in addition to Fielder and CB037. Further, the application of *TaWOX5* in monocots, such as triticale (*Triticum x Secale*), rye (*Secale cereale*), barley (*Hordeum vulgare*), and maize, significantly improved their genetic transformation efficiency, for instance, as observed in maize inbred lines B73 and A188, whose transformation frequencies were significantly enhanced from 19% to 38% [42].

### 2.5. DOF TFs Boost Regeneration Efficiency

DOF (DNA binding with One Finger) proteins represent a plant-specific transcription factor family, initially identified in maize [82] and also in other eukaryotes [83]. *TaDOF5.6* and *TaDOF3.4* can enhance the genetic transformation efficiency in wheat [44]. The efficiency of callus induction by tissues transformed with *TaDOF5.6* and *TaDOF3.4* is increased from 26% to 50% and 55%, respectively (Table 1). Furthermore, *TaDOF5.6* and *TaDOF3.4* also enhance the callus induction rate and genetic transformation efficiency for the wheat varieties KN199 and JM22, which were originally difficult to regenerate [44]. In tissues transformed with *TaDOF3.4* or *TaDOF5.6*, the expression levels of the root and shoot meristematic tissue-related genes *LATERAL ORGAN BOUNDARIES DOMAIN 4* (*TaLBD4*) and *SCARECROW-LIKE 27* (*TaSCL27*) are significantly elevated, and TaDOF3.4 directly activates the expression of *TaLBD4* and *TaSCL27* [44] (Figure 2). Hence, *TaDOF5.6* and *TaDOF3.4* are important DRs in promoting genetic transformation efficiency in wheat. However, there are no reports on *DOF* applications for genetic transformation in maize nor other crops.

### 2.6. LBD Transcription Factors Are Key Regulators of Auxin-Induced Callus Formation

The *LBD*/*ASL* (*ASYMMETRIC LEAVES2-LIKE*) gene family, with a highly conserved *AS2*/*LOB* (*ASYMMETRIC LEAVES2/LATERAL ORGAN BOUNDARIES*) domain at its N-terminus, activates lateral root development to promote the formation of callus tissue [84,85,86]. Auxin response factors ARF7 and ARF19 regulate four downstream *LBD* genes, namely *LBD16*-*18* and *LBD29*, which influence lateral root development and induce callus tissue formation [84]. Among them, LBD16 can directly activate *FAD-binding berberine* (*FAD-BD*), thereby promoting the formation of callus tissue [87]. In *Arabidopsis*, LBD18 activates *EXPANSIN14* (*EXP14*) by directly binding to its specific promoter elements. *EXP14* encodes a cell wall-loosening factor and promotes the occurrence of lateral roots [88] (Figure 2). This indicates that *LBDs* are involved in promoting cell proliferation during the formation of callus tissue. In cotton, the high level of expression of *GhLBD18* is positively correlated with larger callus tissues and significantly accelerates the proliferation rate of callus tissue [89]. This process is effectively driven by the β-estradiol-inducible promoter *pER8*, thus offering a new strategy to replace traditional selective marker systems and provide new possibilities for enhancing cotton transformation efficiency and the genetic improvement capacity (Figure 2).

### 2.7. Overexpression of GOLDEN2 Promotes the Differentiation of Callus Tissue

GOLDEN2 (G2) is a member of the GARP transcription factor family, which includes ARR-B and Psr1 [90,91]. G2 contains the TEA domain and controls the development of chloroplasts and other biological processes [92]. Recently, the expression of *rZmG2* in rice, which is codon optimized based on the maize *G2* gene sequence, significantly improved the genetic transformation efficiency in rice [43]. Mature seeds of three rice varieties and the IE of a maize variety were used in this study to assess the role of the *rZmG2* gene on the genetic transformation efficiency. As a result, transgenic maize and rice callus tissues exhibited early chloroplast formation and increased chlorophyll content during the differentiation stage, with a significant increase in the regeneration efficiency from 26% to 178.46% compared to wild-type callus tissue (Table 1). Further studies have revealed that chloroplast development-related genes, such as *chlorophyll a/b binding protein 24* (*OsCP24*) and *light harvesting chlorophyll a/b binding* (*OsLhcb2* and *OsLhca4*), are significantly upregulated in rice callus tissues carrying the *rZmG2* gene. Additionally, the expression of plant hormone-related genes (e.g., *small auxin-up RNA8 OsSAUR8*) and late embryogenesis abundant (LEA) protein-encoding genes (*OsLEA3*, *14*-*16*, and *22*) are also affected [43] (Figure 2). Notably, the high level of expression of *OsSAUR8* in transgenic callus tissues can promote differentiation by lowering the auxin concentration, as the differentiation of callus tissues requires a low ratio of auxin to cytokinin. During the genetic transformation of rice, the desiccation of callus tissues increases their regeneration potential. Thus, the low expression of *LEAs* in *rZmG2* transgenic callus tissues is advantageous for reducing water retention, which can promote callus tissue desiccation and accelerate regeneration [43].

### 2.8. GRF–GIF Chimeras Enhance Plant Regeneration Efficiency

*GRFs* encode proteins containing QLQ (glutamine–leucine–glutamine) and WRC (tryptophan–arginine–cysteine) domains, which represent a class of plant-specific transcription factors [33]. The QLQ domain facilitates interactions between proteins and the WRC domain is involved in DNA binding [93]. GRFs interact with growth-regulating factor interacting factors (GIFs) to form a plant-specific transcription activation complex [33]. The GRF–GIF complex promotes organogenesis by activating a series of genes, such as *HECATE1* (*AtHEC1*), *HOMEOBOX PROTEIN 33* (*AtHB33*), *CYTOKININ RESPONSE FACTOR 2* (*AtCRF2*), and *CONSTANS-like 5* (*AtCOL5*) [94] (Figure 2). Additionally, AtGIF can repress the expression of the *PLT1* gene and affect ectopic root formation and normal cotyledon development [94]. In poplar, PpnGRF5-1 can reduce the expression level of *cytokinin oxidase/dehydrogenase 1* (*PpnCKX1*), which leads to the accumulation of cytokinin and, thus, promotes the proliferation and formation of meristematic tissue during leaf development [95].

To evaluate the role of *GRF* and *GIF* genes in the genetic transformation of plants, a GRF4–GIF1 chimera was overexpressed in wheat, using the maize *ubiquitin* promoter [34] (Table 1). This approach achieved an almost 8-fold increase in the transformation efficiency in the wheat variety Kronos, significantly higher than that achieved through expressing *GRF4* or *GIF1* alone. Similarly, the GRF4–GIF1 chimera has also successfully enhanced the transformation and regeneration efficiency in other wheat varieties and the rice variety Kitaake [34]. Its application in sorghum has increased the transformation efficiency to nearly eight times that of the original method, without any significant growth defects [96]. Moreover, the overexpression of *GRF5* in *Arabidopsis* increases the chlorophyll content and delays leaf senescence [97]. The application of *GRF5* and its homologs in beet and maize also significantly improve the genetic transformation efficiency, especially the outstanding transformation results achieved by using the *ZmGRF5-LIKE1* gene in maize. The application of the *GRF5* gene in soybean shows the potential to enhance bud proliferation, but decreases root formation and negatively impacts whole plant regeneration, which thus needs to be further optimized [35]. The role of *GRF5* in enhancing genetic transformation efficiency is significant for sugar beet (*Beta vulgaris*), canola (*Brassica napus*), and sunflower (*Helianthus annuus*), with the most pronounced effect on sugar beet [35]. Overall, GRF4–GIF1, GRF5, and their homologs can effectively enhance the regeneration of monocotyledonous plant species (Table 1). However, the specific pathways and molecular mechanisms involved remain undefined.

### 2.9. REF1 Peptide Regulates the Process of Plant Regeneration

Recent research has identified a novel regeneration factor, REGENERATION FACTOR1 (REF1), derived from the *SPR9* gene in tomato (*Solanum lycopersicum*) [45]. *SPR9* encodes a precursor protein for the small peptide SlPep, a member of the plant elicitor peptide (Pep) family, consisting of 23 amino acids. Plant elicitor peptides (Peps), such as SlPep, function as damage-associated molecular patterns (DAMPs) that play critical roles in the plant immune system [98,99,100]. These peptides are released in response to damage, activating defense mechanisms by binding to specific receptors on plant cells known as Pep receptors (PEPRs). This binding triggers a cascade of defense responses that enhance the resistance to pathogens, including bacteria, fungi, and herbivores [100].

Beyond their role in immunity, Peps have also been found to influence plant regeneration processes. The study demonstrated that *SPR9* is crucial for regeneration in tomatoes. Knocking out *SPR9* resulted in the loss of wound-induced callus formation and organ regeneration capacity, whereas the overexpression of *SPR9* or the exogenous application of the peptide significantly enhanced the plant’s regeneration ability. These findings indicate that SlPep not only functions as a key signaling molecule in immune responses, but also plays a pivotal role in promoting regeneration. Consequently, this peptide has been renamed REGENERATION FACTOR 1 (REF1) to reflect its dual function in both defense and regeneration, underscoring its importance as an endogenous signaling molecule in plant responses to damage and stress. Further research has confirmed that REF1 exerts its effects through binding to the leucine-rich repeat receptor-like kinase PORK1 (PEPR1/2 ORTHOLOG RECEPTOR-LIKE KINASE 1), which serves as its receptor. Upon the recognition of REF1 by PORK1, the expression of the key downstream regulator *SlWIND1* is activated (Figure 2). *SlWIND1* is crucial for cellular reprogramming, initiating tissue repair and organ regeneration processes (Figure 2). Additionally, SlWIND1 binds to the promoter region of the *REF1* precursor gene, enhancing its expression and leading to the production of more REF1 peptides, thereby amplifying the REF1 signal [45].

Importantly, the function of REF1 is conserved across the plant kingdom. REF1 peptides and their receptors are found in nearly all dicots and monocots. The exogenous application of REF1 significantly increased the transformation efficiency of wild tomato species (*S. peruvianum* and *S. habrochaites*) by 6 to 12-fold. Similarly, in soybean, the application of *GmREF1* improved the transformation efficiency of the Dongnong-50 variety by 2.5-fold, demonstrating its potential to promote gene integration and regeneration in soybean cells. In wheat, *TaREF1* was applied to immature embryos, resulting in a 4-fold increase in transformation efficiency in the JM22 variety, highlighting its crucial role in promoting regeneration in Gramineae plants. Similarly, in maize, the expression of *ZmREF1* increased the transformation efficiency of the B104 variety by 4-fold. Collectively, these results indicate that *REF1* can effectively enhance transformation efficiency across diverse plant species [45] (Table 1), suggesting its broad applicability in the fields of plant biotechnology and bio-breeding.

### 2.10. Overexpression of LAX1 Enhances Shoot Regeneration Efficiency in Plants

TaLAX1 is a recently identified developmental regulator, whose overexpression has been shown to significantly increase the regeneration efficiency of the wheat genotype Fielder [68]. Prior research has revealed that loss-of-function mutations in *TaLAX1* result in a compact spike phenotype [101]. *LAX PANICLE1* (*OsLAX1*) and *BARREN STALK1* (*ZmBA1*), the homologous genes of *TaLAX1* in rice and maize, respectively, encode atypical basic helix–loop–helix transcription factors [102,103]. *OsLAX1* is critical for the formation of axillary meristems in rice, while *zmba1* mutants exhibit unbranched and shortened tassels. However, the involvement of these genes in plant regeneration remains unclear.

Further investigations have demonstrated that the overexpression of *TaLAX1-A* promotes shoot regeneration across a range of wheat genotypes, including five common wheat varieties (Chinese Spring, Kenong 199, Shannong 28, Aikang 58, and Jimai 22) and two durum wheat varieties (Langdon and Kronos), including those known to be recalcitrant to transformation [68]. RNA-sequencing (RNA-seq) analysis of Chinese Spring calli, conducted five days after the transfer to a shoot-induction medium, revealed that tissues overexpressing *TaLAX1-A* exhibited upregulation of several key genes, including *ETHYLENE RESPONSE FACTOR 115* (*ERF115*), *WIND1*, *ESR1*, *PLT3*, *PLT5*, and *PLT7*. Additionally, endogenous levels of hormones, such as auxin, cytokinin, and jasmonic acid, were significantly elevated in calli overexpressing *TaLAX1-A* compared to the controls. *TaLAX1-A* was also found to directly regulate the expression of *TaGIF1*. These findings suggest that *TaLAX1-A* may regulate shoot regeneration through a mechanism akin to the TaGRF4–TaGIF1 pathway, indicating a regulatory network involving auxin, cytokinin, jasmonic acid, and related transcription factors, like *TaGRFs* and *TaGIF1* (Figure 2), which are integral to cell division, cell fate transition, and shoot regeneration.

In addition to elucidating the role of *TaLAX1* in wheat regeneration, researchers have explored potential applications of its homologous genes in soybean and maize (Table 1) [68]. Notably, the overexpression of *GmLAX1* in soybean variety Dongnong-50 resulted in an approximate 1.7-fold increase in regeneration efficiency, highlighting its potential for promoting soybean tissue culture and regeneration. Conversely, the overexpression of *ZmBA1* in maize variety B104 led to an approximate 3.4-fold increase in the regeneration efficiency, demonstrating that *ZmBA1* significantly enhances cell division and regeneration capabilities in maize [68].

## 3. Conclusions and Perspectives

The sequential stages of plant regeneration, from explants to whole plant formation, involve various DRs that play distinct roles during each phase. Understanding these stages and the specific contributions of different DRs provides a framework for optimizing genetic transformation processes. Figure 3 illustrates these stages, highlighting how individual DRs contribute to specific phases of plant regeneration, thereby improving the overall efficiency of genetic transformation (Table 1).

At the initial stage of co-cultivation, specific DRs, such as LBD, rZmGOLDEN2, and REF, are crucial for promoting the dedifferentiation of plant cells, ensuring robust callus formation. As the callus develops, DRs like SERK, WIND1, WOX, and PLT, become essential for maintaining embryogenic competence and guiding tissue organization, accelerating somatic embryo formation. Finally, DRs, such as GRF–GIF complexes and LAX1, facilitate shoot regeneration by regulating cell proliferation and auxin gradients, ensuring the development of fully functional plants (Figure 3).

These observations suggest that a strategic, stage-specific application of DRs maybe significantly enhance transformation efficiency by tailoring the combination of DRs to the specific needs of each regeneration stage. This approach may address genotype dependency and increase the success rates of transformation, supporting the development of healthy plants from explants to maturity.

A holistic approach to employing DRs, informed by a deeper understanding of their roles at each stage, can help design transformation protocols adaptable to a wide range of plant species and genotypes. Future research should prioritize optimizing DR expression levels, timing, and refining their combinations, while also identifying novel DRs to enhance transformation efficiency. Integrating these insights with cutting-edge technologies, such as CRISPR-based gene editing, transgenic breeding, and single-cell RNA sequencing, could provide deeper molecular insights, leading to more robust strategies for crop development.

Furthermore, addressing regulatory and safety concerns associated with gene-edited and transgenic plants is crucial. Developing strategies for monitoring gene flow, assessing environmental impacts, and identifying potential health risks, will ensure the responsible application of these technologies. Innovative approaches, like trait stacking, should also be explored to improve complex agricultural traits and address diverse environmental challenges [3,15].

In conclusion, the strategic application of multiple DRs across different stages of plant regeneration, combined with technological advancements, will provide a comprehensive approach to overcoming the current limitations related to plant genetic transformation. This integrated strategy not only enhances our understanding of plant developmental biology, but also offers practical solutions for sustainable agriculture and food security. By addressing both the scientific and regulatory challenges, we can pave the way for more resilient and sustainable agricultural practices.

## Figures and Tables

**Figure 1 plants-13-02841-f001:**
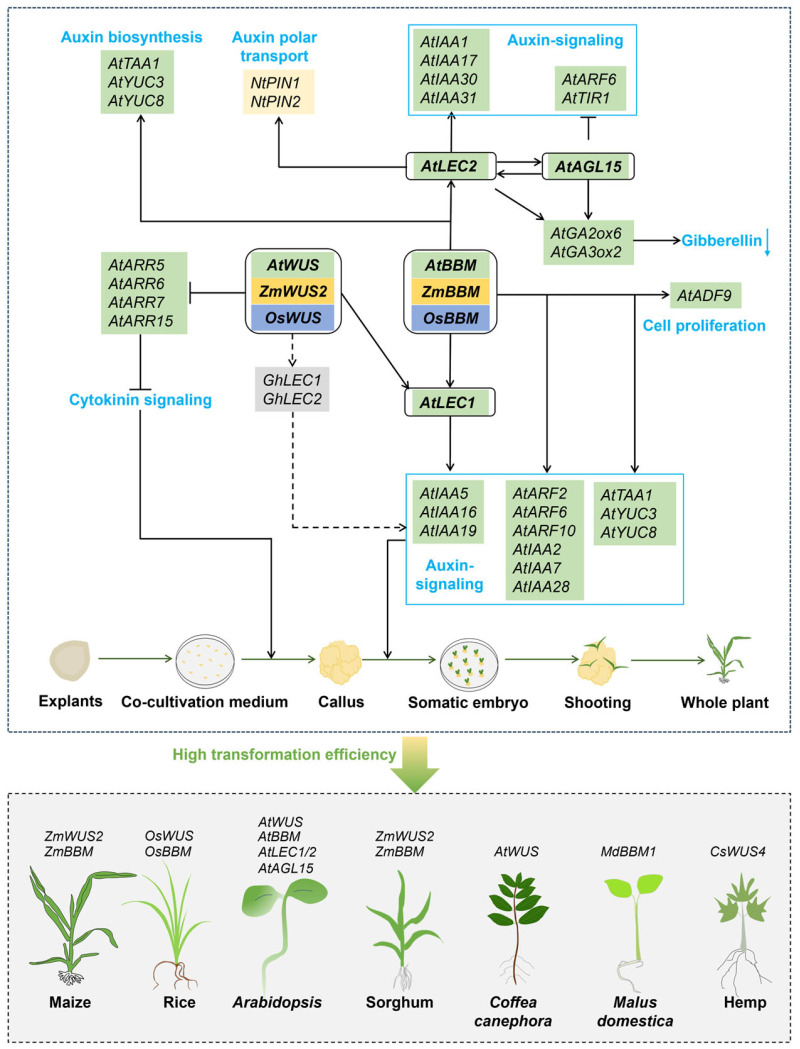
Schematic representation of *WUS* and *BBM* genes and their roles in enhancing plant genetic transformation. Genes on the green background are from *Arabidopsis*, those on the yellow background are from maize, those on the blue background are from rice, those on the dark gray background are from cotton (*Gossypium*), and those on the faint yellow background are from tobacco (*Nicotiana tabacum*). The light blue font and boxes indicate hormone-related signaling pathways and biological processes, as well as their related genes. The genes above the different plants denote that they have been employed to improve the genetic transformation efficiency in the respective plant species. The co-cultivation medium is primarily used for the *Agrobacterium* infection process and is not involved in the callus induction in explants. The processes illustrated reflect maize transformation, as an example.

**Figure 3 plants-13-02841-f003:**
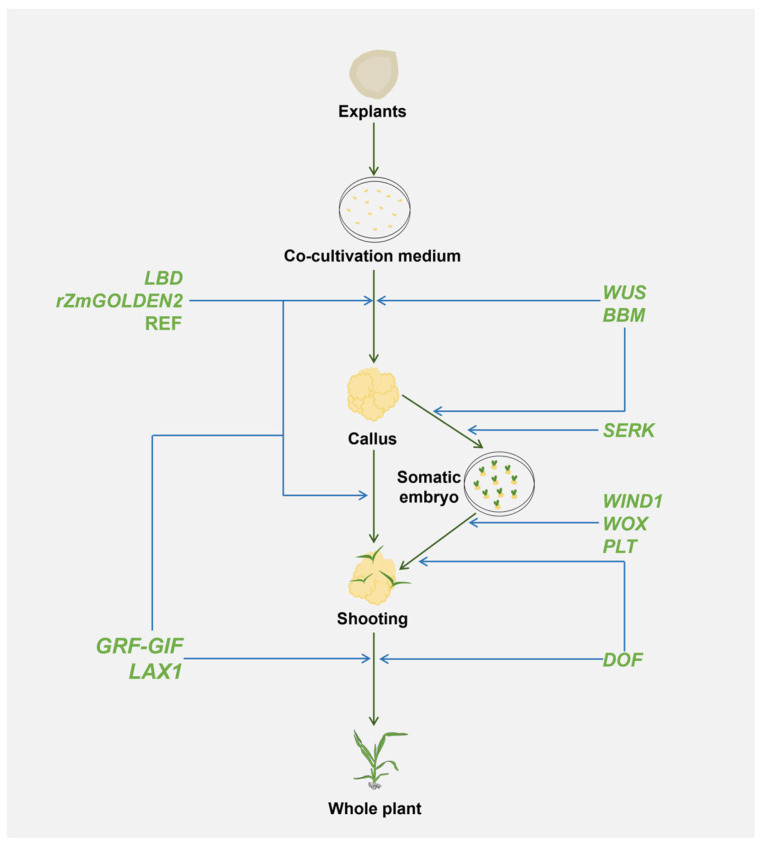
Schematic representation of developmental regulatory factors that play key roles in the different stages of the whole plant regeneration. The co-cultivation medium is primarily used for the *Agrobacterium* infection process and is not involved in the callus induction in explants. The processes illustrated reflect maize transformation, as an example.

## Data Availability

Data are contained within the article.

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
