# Peer review of "Functional Mechanisms and the Application of Developmental Regulators for Improving Genetic Transformation in Plants"

_plants, 2024, doi:10.3390/plants13202841_

Round 1
Reviewer 1 Report
Comments and Suggestions for Authors
Dear the editor,
The submitted manuscript entitled “Functional mechanism and application of developmental regulators for improving genetic transformation in plants” describes the genes used for improving the efficiency in plant transformation, the mechanisms regulated by those genes and application methods including the transformation vector systems.
Because the manuscript covers a lot of genes and crop species, this paper would be informative for a wide range of readers.
To further improve the manuscript, I would like to suggest some points below.
1. In figures, “co-cultivation medium” are drawn. The co-cultivation medium is used for the process of agrobacteria infection, but the figures sometimes seem to represent the process of callus induction from explants. It should be better to clarify the two processes.
2. In Figure1, the rice WUS gene, OsWUS is drawn. However, involvement of OsWUS in the mechanism of transformation is not described in the manuscript at all.
3. In Figure1, tabacco genes NtPIN1 and NtPIN2 are drawn. The explanation on these genes should be added in the text.
4. In Figure2, PROK1 is typo.
5. In Line 112, Coffea canephora is described as “coffea”, but in Line 173 “coffee plants” and in Line 190 “coffee” are used respectively. The authors should use same word throughout the manuscript.
Author Response
Comments 1: In figures, “co-cultivation medium” are drawn. The co-cultivation medium is used for the process of agrobacteria infection, but the figures sometimes seem to represent the process of callus induction from explants. It should be better to clarify the two processes.
Response 1: Thank you for your feedback regarding the clarification of the two processes in our figures. In the revised manuscript, we have modified the figure legends for all three figures to specify that the co-cultivation medium is used primarily for the Agrobacterium infection process and is not involved in callus induction.
Comments 2: In Figure1, the rice WUS gene, OsWUS is drawn. However, involvement of OsWUS in the mechanism of transformation is not described in the manuscript at all.
Response 2: Thank you for your valuable suggestion. we have added a description of the potential relationship between OsWUS and the transformation mechanism in lines 83-84 of the revised manuscript. This addition clarifies how OsWUS may contribute to the process and enhances the overall understanding of its significance in relation to genetic transformation.
Comments 3: In Figure1, tabacco genes NtPIN1 and NtPIN2 are drawn. The explanation on these genes should be added in the text.
Response 3: Thank you for your valuable suggestion. we have added a description of these genes and their roles in the context of auxin polar transport in lines 98-99 of the revised manuscript. This addition will provide clarity and enhance the understanding of their significance in relation to the transformation process.
Comments 4: In Figure2, PROK1 is typo.
Response 4: Thank you for pointing out the typo regarding PROK1 in Figure 2. We have corrected this error in the images of the revised manuscript.
Comments 5: In Line 112, Coffea canephora is described as “coffea”, but in Line 173 “coffee plants” and in Line 190 “coffee” are used respectively. The authors should use same word throughout the manuscript.
Response 5: Thank you for the kind comment. we have consistently used the term "coffee plants" throughout the revised manuscript, specifically making these changes in lines 112 and 192. This adjustment enhances clarity and maintains uniformity in our descriptions.
Reviewer 2 Report
Comments and Suggestions for Authors
In the submitted manuscript, the authors investigate the possibility of enhancing the genetic transformation efficiency by molecular mechanisms and regulatory networks of many developmental regulators. This review article gives a slightly different but interesting overview on genetic transformation of plants. It is known that successful genetic transformation depends on plant genotype, regeneration efficiency and transformation process itself. This review suggests that strategic, stage-specific application of developmental regulators might significantly enhance the genetic transformation efficiency. This a slightly different approach, may be possible to address genotype dependency and increase the success rates of genetic transformation, supporting the development of healthy regenerated plants. The authors presented well-known genes (SERK, LEC, BBM, WUS etc.) in a completely different context and explained that modulating the expression of these genes the significant improvement in the rate of genetic transformation can be make. The manuscript is also supplement with illustrative and appropriate schematic presentations.
Author Response
Comments:In the submitted manuscript, the authors investigate the possibility of enhancing the genetic transformation efficiency by molecular mechanisms and regulatory networks of many developmental regulators. This review article gives a slightly different but interesting overview on genetic transformation of plants. It is known that successful genetic transformation depends on plant genotype, regeneration efficiency and transformation process itself. This review suggests that strategic, stage-specific application of developmental regulators might significantly enhance the genetic transformation efficiency. This a slightly different approach, may be possible to address genotype dependency and increase the success rates of genetic transformation, supporting the development of healthy regenerated plants. The authors presented well-known genes (SERK, LEC, BBM, WUS etc.) in a completely different context and explained that modulating the expression of these genes the significant improvement in the rate of genetic transformation can be make. The manuscript is also supplement with illustrative and appropriate schematic presentations.
Response: We are pleased to learn that you find this review article provides an interesting perspective on the genetic transformation of plants. We also appreciate your acknowledgment of our suggestion regarding the potential for strategically applying developmental regulators to significantly enhance genetic transformation efficiency. We appreciate your support and recognition of our work, and we look forward to your further review.